# Single Cell Effects of Photobiomodulation on Mitochondrial Membrane Potential and Reactive Oxygen Species Production in Human Adipose Mesenchymal Stem Cells

**DOI:** 10.3390/cells11060972

**Published:** 2022-03-11

**Authors:** Li-Chern Pan, Nguyen-Le-Thanh Hang, Mamadi M.S Colley, Jungshan Chang, Yu-Cheng Hsiao, Long-Sheng Lu, Bing-Sian Li, Cheng-Jen Chang, Tzu-Sen Yang

**Affiliations:** 1Graduate Institute of Biomedical Optomechatronics, Taipei Medical University, Taipei 110, Taiwan; lcpan@tmu.edu.tw (L.-C.P.); m850109005@tmu.edu.tw (N.-L.-T.H.); ms_colley81@hotmail.com (M.M.C.); ychsiao@tmu.edu.tw (Y.-C.H.); jacob666j@alumni.ncu.edu.tw (B.-S.L.); 2Graduate Institute of Medical Sciences, College of Medicine, Taipei Medical University, Taipei 110, Taiwan; js.chang@tmu.edu.tw; 3Graduate Institute of Biomedical Materials and Tissue Engineering, Taipei Medical University, Taipei 110, Taiwan; lslu@tmu.edu.tw; 4International Ph.D. Program in Biomedical Engineering, Taipei Medical University, Taipei 110, Taiwan; 5Department of Radiation Oncology, Taipei Medical University Hospital, Taipei Medical University, Taipei 110, Taiwan; 6Department of Medical Research, Taipei Medical University Hospital, Taipei 110, Taiwan; 7TMU Research Center of Cancer Translational Medicine, Taipei Medical University, Taipei 110, Taiwan; 8Center for Cell Therapy, Taipei Medical University Hospital, Taipei Medical University, Taipei 110, Taiwan; 9Department of Plastic Surgery, Taipei Medical University Hospital, Taipei 110, Taiwan; 10Department of Surgery, College of Medicine, Taipei Medical University, Taipei 110, Taiwan; 11School of Dental Technology, Taipei Medical University, Taipei 110, Taiwan; 12Research Center of Biomedical Device, Taipei Medical University, Taipei 110, Taiwan

**Keywords:** photobiomodulation, human adipose-derived mesenchymal stem cell, mitochondrial membrane potential, reactive oxygen species, vesicle transport

## Abstract

Photobiomodulation (PBM) has recently emerged in cellular therapy as a potent alternative in promoting cell proliferation, migration, and differentiation during tissue regeneration. Herein, a single-cell near-infrared (NIR) laser irradiation system (830 nm) and the image-based approaches were proposed for the investigation of the modulatory effects in mitochondrial membrane potential (ΔΨm), reactive oxygen species (ROS), and vesicle transport in single living human adipose mesenchymal stem cells (hADSCs). The irradiated-hADSCs were then stained with 2′,7′-dichlorodihydrofluorescein diacetate (H_2_DCFDA) and Rhodamine 123 (Rh123) to represent the ΔΨm and ROS production, respectively, with irradiation in the range of 2.5–10 (J/cm^2^), where time series of bright-field images were obtained to determine the vesicle transport phenomena. Present results showed that a fluence of 5 J/cm^2^ of PBM significantly enhanced the ΔΨm, ROS, and vesicle transport phenomena compared to the control group (0 J/cm^2^) after 30 min PBM treatment. These findings demonstrate the efficacy and use of PBM in regulating ΔΨm, ROS, and vesicle transport, which have potential in cell proliferation, migration, and differentiation in cell-based therapy.

## 1. Introduction

Stem cell therapies have emerged in recent years as a means to provide potential treatment for patients due to their safety and effectiveness compared to traditional therapies. There are several reasons why stem cells are widely used as an ideal candidate for medical treatments for tissue repair. The most important reason is that stem cells have intrinsic properties, such as self-renewal and pluripotential properties, which can facilitate the regeneration of different cell types [1,2]. Nonetheless, mesenchymal stromal cells (MSCs) appear to be a preferred candidate, especially the adipose stem cells (ADSCs), are rich in sources and can be readily harvested from adult human adipose tissue with minimum pain and invasiveness, along with higher progenitor differentiation capacity, which includes osteoblasts, adipocytes, chondrocytes, hepatocytes and neurocytes [3,4,5].

Photobiomodulation (PBM) of human adipose-derived mesenchymal stem cells (hADSCs) using near-infrared (NIR) laser irradiation has recently become more popular as an auxiliary treatment for cell therapies. This is because PBM can result in the acceleration of stem cell proliferation in vitro and migration in vivo conditions [6,7,8]. Nevertheless, the underlying mechanism remains unclear for the above-mentioned efficacies. Among the most well-known proposed mechanisms of PBM, cytochrome-c oxidase (CCO) plays a role as a major photoreceptor in the mitochondria [9,10]. Deeply, CCO is located at unit IV of the mitochondrial respiratory chain (Figure 1) and be responsible for transferring four protons from cytochrome c to one molecular oxygen to generate two water molecules. This translocation of four protons was occurred across the mitochondrial membrane to synthesize adenosine triphosphate (ATP) [11,12]. Another mechanism regarding cytochrome c was reported that mitochondria can use cytoplasmic cytochrome c to maintain mitochondrial transmembrane potential and ATP production after outer membrane permeabilization [9], which indicated the correlation of mitochondrial membrane potential (ΔΨm) and the roles of cytochrome c in PBM.

On the other hand, it is believed that the more ΔΨm was generated, the more ROS was produced [13]. Previous studies have pointed out that the reactive oxygen species (ROS) level and ΔΨm are positively correlated [14,15]. For instance, the rate of ROS production has been shown to increase significantly when mitochondria transmembrane potential ΔΨm is above 140 mV and decreases when the ΔΨm level drop down to 10 mV [16]. Nevertheless, another study demonstrated an excessive increase in ROS production could lead to a concomitant decrease in ΔΨm due to “ROS-induced ROS release” (RIRR), which is a regenerative cycle of mitochondrial ROS formation and release through mitochondrial permeability transition pore (mPTP). Since under prolonged oxidative stress, sustained mPTP opening will lead to a sudden bust of ROS that causes the destruction of mitochondria along with a decrease in ΔΨm [17]. In addition, the increased ΔΨm and decreased ROS production could occur in some cases due to a phenomenon of “proton leak”, which refers to the mitochondrial respiration proportion in the presence of ATP synthase inhibitor Oligomycin [18]. From that, it evokes some questions regarding the fluence-dependent of PBM in promoting ROS and ΔΨm; and the correlation between them.

Indeed, there are millions of biological processes in the single living cell at any given instantaneous, including enzymatic reactions, polymerization, and depolymerization, transportation of cell migration, or cytoskeletal rearrangement [19]. Thus an additional investigation in vesicle transport regarding the mechanical properties of the cytoskeleton of stem cells was conducted in this single-cell research, which impacts cell movement and adhesion [20]. Furthermore, vesicle transport is related to the cellular cytoskeleton that can convert chemical energy in ATP synthesis into mechanical motion [19]. Thus, the spontaneous fluctuation in vesicle transport after PBM is confirmed by measuring the velocity of vesicle motion to answer whether the increase of ΔΨm ad ROS is correlated with the increased vesicle transport or not.

Therefore, this single-cell research mainly investigated the changes of the ΔΨm and ROS and vesicle transport in the irradiated-hADSCs at a range of fluence 2.5–10 (J/cm^2^). Additionally, the time-lapse fluorescence microscopy was conducted to identify an appropriate time point for recording the responses of the ΔΨm and ROS after PBM. From that, one of the best fluence from 2.5 to 10 J/cm^2^ can be identified for the ΔΨm and ROS promotion, which might link to the potential cellular functions, including stem cell proliferation, migration, and differentiation [17,21].

## 2. Materials and Methods

### 2.1. Experimental System

The inverted microscopy (TE2000U, Nikon, Japan) is equipped with an objective lens (Plan Apo 90 ×/1.40 oil) and two types of CCD cameras, including an electron-multiplying CCD (EMCCD) camera (Luca^EM^ S658M, Andor, Belfast, UK) and a sCMOS camera (ORCA-Flash4.0 V3, Hamamatsu Photon, Hamamatsu, Japan) (Figure 2A). First, an 830 nm laser system (Lambda Beam Wavelock, RGB Photonics, Bavaria, Germany) was passed over plano-convex lenses of focal lengths (L1 and L2), with f = 2.5 cm and f = 10 cm, respectively (Figure 2A), namely, the output diameter was four times larger than that of the diameter of the input beam. Subsequently, followed by passing through another plano-convex lens of local lengths of L3 (f = 12.5 cm) and L4 (f = 25 cm), the diameter of the laser beam was further increased twofold. Therefore, as a result, there were an eightfold increase in diameter of the input beam (Figure 2A). After the above process, the laser beam was then reflected by the dichroic mirror D1 (780dcspxr, Chroma, Irvine, CA, USA) on the inverted microscope and passes through the condenser into a microfluidic flow chamber for single hADSC NIR laser irradiation. The red dotted box demonstrates the operation mode for the single-cell NIR laser irradiation, where the laser spot area, either 300 µm^2^ or 600 µm^2^, can cover the whole single hADSC. The laser power was measured and calibrated by the ophir power meter (PD300-3W-V1 and Starlite, Jerusalem, Israel). Figure 2B showed the experimental setup for the proposed single-cell PBM irradiation system.

### 2.2. Cell Culture and Chemicals

Human mesenchymal adipose stem cell line (HMSC-ad 7510) used for this study was provided by ScienCell Research Laboratories (Carlsbad, CA, USA). Briefly, the 10^6^ viable hADSCs were then thawed and placed in human mesenchymal-LS expansion medium (MilliporeSigma, Burlington, NJ, USA) containing 5 ng/mL rhFGF-b, 50 µg/mL Ascorbic acid, 7.5 nM L-glutamine, 1.0 µg/mL Hydrocortisone Hemisuccinate, 5 µg/mL rh Insulin and 10% fetal bovine serum (FBS) at 37 °C and 5% CO_2_ until the cells are approximately 80% confluent. The subculture process was carried out continuously with an accutase solution (MilliporeSigma) until the required number of hADSCs was sufficient of the 80% confluency. Specifically, these detached hADSCs were transferred to a 15 mL conical tube and centrifuged at 300× *g* for 3–5 min. From the pellet, the cells were then resuspended in human mesenchymal-LS expansion medium and finally seed into the slide chamber for further experiments at a density of 2 *×* 10^5^ cells/mL at 37 °C and 5% CO_2_ within 24 h [22,23,24].

### 2.3. Time-Lapse Rhodamine 123 Probe of Mitochondrial Membrane Potential in PBM

In this study, the investigation of the time-lapse Rh123 probe of the ΔΨm was conducted to locate a suitable point in time for the induction of the mitochondrial membrane potential after PBM that triggers cell proliferation and migration [8,25,26,27,28]. The Rh123 dye’s best concentration (5 µM) was tested within a range of dye concentrations (1, 5, 10 µM) that was shown in (Appendix A). In addition, there are three main control factors: treatment doses (J/cm^2^), real power, and treatment time. Nevertheless, the area and real power in the chamber were kept at a constant level of 3 × 10^−^^6^ cm^2^ (Appendix A) and 0.076 W (Appendix A), respectively. Meanwhile, various active power in the chamber was implemented, including a wide range of time points of 0, 5, 10, 20, 30, 60 (min) to deliver corresponding doses at 2.5, 5, and 10 (J/cm^2^), respectively (Appendix A). The real power that the irradiated-hADSCs were absorbed was calculated based on the power consumption and 59% transmission of Nikon microscope objectives (Plan Apo CFI) for the wavelength 830 nm, and the area can then be estimated depending on a specific investigated dose. Since the higher the fluorescent signal intensity of the Rh123 expression, the higher ΔΨm is [29,30,31], therefore, we defined the Rh123’s mean fluorescence intensity ratio (S_2_/S_1_) before (S_1_) and after (S_2_) PBM to quantify the changes of ΔΨm for the hADSCs irradiated with 830 nm PBM at the single-cell level.

### 2.4. Detection of Mitochondrial Membrane Potential in PBM

The ΔΨm was detected using the Rhodamine 123 fluorescence probe (Promocell GmbH, Heidelberg, Germany) to label mitochondrial membrane potential. Initially, 2 × 10^5^ (cells/mL) of hADSCs were first seeded into a microfluidic cell culture platform for 24 h at 37 °C in 5% CO_2_. The hADSCs were then maintained with the Rh123 dye (5 µM) for mitochondrial membrane staining for 30 min, followed by PBS (1X) washing once. Subsequently, the hADSCs were exposed to laser irradiation (830 nm, 0.076 W), with various fluence of 2.5, 5, and 10 J/cm^2^ where the laser exposure times are 5, 10, 20 s, respectively. The responses of the hADSCs were recorded after 30 min with an exposure time of 0.5 s via a laser system (wavelength 830 nm) (Figure 2). The change of ΔΨm was expressed as the Rh123’s mean fluorescence intensity ratio before and after PBM. The increase of the fluorescence signals of Rh123 probe illustrated mitochondrial membrane depolarization [32].

### 2.5. Assessment of ROS in PBM

The intracellular reactive oxygen species (ROS) level was monitored using the ROS sensitive probe (2′,7′-dichlorodihydrofluorescein diacetate, H_2_DCFDA, Invitrogen, D-399). To make a 10 mM stock solution, H_2_DCFDA was diluted in Dimethyl sulfoxide (DMSO, MedChemExpress, Monmouth Junction, NJ, USA, HY-15534) and then further diluted before use. Adherent cells (Human Adipose Stem Cells, ADSC) in a flow chamber were stained for 30 min in the dark at room temperature with a 5 µM staining solution in PBS. Then, using an 830 nm infrared diode laser, the control and PBM-exposed cells were examined at various flux intensities. The H_2_DCFDA staining method was the same as the H_2_DCFDA staining method. The laser irradiation protocol is followed as in Appendix A. At fluences of 2.5, 5, and 10 J/cm^2^), intracellular ROS was expressed as a ratio (R_2_/R_1_) before (R_1_) and after (R_2_) 30 min of PBM.

### 2.6. Vesicle Transport

To probe the intracellular response of hADSCs before and after 30 min PBM, the EMCCD camera was used to acquire bright-field images at a frame rate of 7.87 frames per second (fps) for imaging the dynamics of intracellular vesicles within the living cell. Furthermore, Matlab-based FFT software (Matlab R2021a) for cross-correlation processing tracks and determines intracellular vesicles’ dynamics in a 64 × 64 pixels interrogated window. Hence, from a total of 100 images, the time series of two-dimensional (2-D) displacements and the corresponding velocities of intracellular vesicles inside the interrogated widow could be obtained and identified. The average of the absolute value of velocity (V) for each interrogated window and the ensemble average of V from individual hADSCs (*n* = 8) can be determined. Here, we define V_1_ and V_2_ as the ensemble average of the velocity before and after PBM, respectively. Finally, we apply the ensemble average of the velocity ratio (V_2_/V_1_) to quantify the intracellular response of hADSCs before and after 30 min PBM at the best fluence (5 J/cm^2^) compared to a control group (0 J/cm^2^).

### 2.7. Imaging Analysis

The use of an exposure time of 0.5 s for fluorescent images and 0.1 s for bright field images was applied in this study. For investigations regarding to ΔΨm, image acquisition and analysis were performed using Matlab software(Matlab R2021a) Deeply, Matlab was used to remove the background noise from fluorescence images (Appendix A), followed by calculating the mean fluorescence intensity of the Rh123 before and after PBM. On the other hand, Matlab was further applied for vesicle transport analysis. Initially, a region of interest (ROI), which is located around the nucleus, was identified to track the changes in vesicle transport. The positions of ROI were then determined by ImageJ software (1.8.0) and were cut with 64 *×* 64 pixels by Matlab software (Matlab R2021a). Followed by measuring the 2-D displacements and the corresponding velocities before and after 30 min PBM (Figure 3).

### 2.8. Statistical Analysis

All data were analyzed and expressed as the mean ± standard error of the mean (SEM) for various independent experiments using GraphPad Prism 8.3.0. One-way analysis of variance (ANOVA) with Tukey HSD (honestly significant difference) post-hoc test was used to assess differences among the groups; *p* < 0.05 and *p* < 0.01 were considered statistically significant and statistically highly significant, respectively. In addition, the effects of PBM on vesicle transport was evaluated using the paired *t*-test.

## 3. Results

### 3.1. Time-Lapse Rh123 of Mitochondrial Membrane Potential in Photobiomodulation

The approach first used the Rh123 probe of 0.5 mM to mark the hADSCs’s mitochondrial membrane. These hADSCs were treated with low, medium and high fluences and captured continuously with various time points (0, 5, 10, 20, 30, 60 min) (Appendix A). The ratio of the mean fluorescence intensity of Rh123 before and after PBM represents the changes in ΔΨm. An appropriate time point was identified by comparing the change in ΔΨm at a wide range of different timepoint. Figure 4 showed that the increased ΔΨm at a time point of 20 min had a significant difference compared to 0 min at the fluences of 0 and 5 (J/cm^2^) (*p* < 0.05), while it was not at other fluences (*p* > 0.05). For a fluence of 10 J/cm^2^, the rate of ΔΨm was increased statistically at a time point of 30 and 60 min PBM. In addition, there was no significant difference in ΔΨm between 30 and 60 min PBM at all groups (*p* > 0.05). In addition, there was no considerable difference between control group and treatment groups at time point of 30 min PBM (*p* > 0.05) on the irradiated-hADSCs.

### 3.2. Effects of PBM in Promoting the Mitochondrial Membrane Potential in hADSCs

In this study, the procedures of PBM were set to investigate the fluence-dependent effects of the irradiated-hADSCs in ΔΨm (Figure 5A). These results were defined by measuring the ratio of the mean Rh123’s fluorescence intensity before and after 30 min PBM (S_2_/S_1_) based on fluorescence images by Matlab software(Matlab R2021a) (Figure 5B), which results in the different changes of ΔΨm in the irradiated-hADSCs (Passage 10). 

In Figure 6, these results demonstrated that PBM stimulated the statistical change in ΔΨm at fluences of 5 and 10 (J/cm^2^) with the same ratio of S_2_/S_1_ = 1.07 compared to 2.5 J/cm^2^ S_2_/S_1_ = 1.00 (*p* < 0.05). Furthermore, there was statistically no difference between the fluence of 5 and 10 (J/cm^2^), and similarly between control group (0 J/cm^2^) and treatment groups in ΔΨm. Nevertheless, these observations were interpreted due to a huge variety among the responses of the irradiated-hADSCs. From these responses, it is obvious that using the different light energy can impact various biological effects due to PBM [12]. From that, the good performance in ΔΨm were at fluence of 5 and 10 (J/cm^2^), S_2_/S_1_ = 1.07 based on these results (Figure 6).

On the other hand, compared to the previous results in time-lapse investigations in ΔΨm (Figure 4). The above results showed there was no considerable difference among treatment groups at time point of 30 min PBM (*p* > 0.05) on the irradiated-hADSCs. Perhap, because of different kinds of camera to capture fluorescence images in these cases. Therefore, for the following investigation, an EMCCD camera (EMCCD) was applied for capturing fluorescence images to enhance the sensitivity of these images instead of using ORCA-Flash 4.0 V3 Ditigal CMOS camera (Hamamatsu Photon, Hamamatsu, Japan).

### 3.3. Effects of PBM in Promoting ROS in hADSCs

To determine the specificity of the mechanism that leads to large amounts of ROS in the irradiated-hADSCs at fluence of 5 J/cm^2^, the cells were treated with an H_2_DCFDA marker for 30 min. Fluorescence images of the cultured-hADSCs Figure 7 and the bar chats Figure 8 displayed the cells with a high level of intracellular ROS. Maxima rise in fluorescence intensity within the cells was found considerably at 5 J/cm^2^ at 30 min from the three laser fluences (2.5, 5, and 10 J/cm^2^). Both 2.5 and 10 J/cm^2^ demonstrated a decrease in ROS intensity. When PBM was either low or high, it was projected that there would be less benefit. At a modest power of 5 J/cm^2^, 30 min, PBM was expected to increase intracellular ROS with a function of time in the hADSCs. One-way ANOVA analysis was performed in this study. There were highly significant differences between 5 J/cm^2^ compared to two fluences of 2.5 and 10 (J/cm^2^), *p* ≤ 0.0001, while there was no significant difference between 2.5 and 10 J/cm^2^.

### 3.4. Vesicle Transport in hADSCs after PBM

Based on the changes of ΔΨm in the irradiated-hADSCs (Figure 6), one of the good performances in ΔΨm was observed at a fluence of 5 J/cm^2^, S_2_/S_1_ = 1.07. Thus, fluence of 5 (J/cm^2^) was selected to investigate vesicle transport because it is believed that the increased ΔΨm and vesicle transport are correlated. For experimental procedures in Figure 9A, the hADSCs were first seeded into flow chamber slides and incubated 24 h for cell attachment, followed by staining with Rh123 within 30 min before treatment. The hADSCs were then treated with a fluence of 5 J/cm^2^. These bright-field images were photographed before and after 30 min of PBM (exposure time and kinetic length of 0.1 s and 100, respectively). In this study, a specific ROI area (64 × 64 pixels) was selected to investigate the changes in vesicle transport (Figure 9B). Noted that this desired ROI included the hADSCs’ mitochondria (Figure 9B), which refers to a significant source of ROS production [33]. From that, this experiment could demonstrate the relationship among ΔΨm, ROS and vesicle transport in detail.

In this study, the vesicle transport was represented through the ratio of the mean velocity before and after 30 min PBM (V_2_/V_1_). In Figure 10, the results () demonstrated that the velocity of individual cells before (V_1_) and after PBM (V_2_) at control group (0 J/cm^2^) and treatment group (5 J/cm^2^) was V_2_/V_1_ = 1.15 (Figure 10A) and V_2_/V_1_ = 2.48 (Figure 10B), respectively. In addition, the PBM-based fluctuation in the velocity (V_2_) at 5 J/cm^2^ was enhanced significantly in the irradiated-hADSCs, as shown in Figure 10B. Furthermore, regarding the ensemble average of the velocity ratio (V_2_/V_1_), the result indicated a significant difference in V_2_/V_1_ = 1.93 of 5 J/cm^2^, when compared to control group (V_2_/V_1_ = 0.76) (*p* = 0.0117 < 0.05) (Figure 10C). From these observation, a fluence of 5 J/cm^2^ of PBM significantly enhanced the ΔΨm and vesicle transport in hADSCs.

## 4. Discussion

Though PBM has widely been emerged in cell-based therapy in recent times [34,35,36]. However, the intracellular efficacy induced by PBM has not yet been revealed clearly. Thus, limits its clinical application in regenerative medicine. Therefore, strategies aiming at improving the cell viability, proliferation, differentiation, as well as cell mobility, will be crucial for further survival of the cell engraftment. In addition, during light irradiation, there are dose-dependent biphasic response observed both in in vitro and animal studies [37,38]. Thus, selection of an appropriate irradiation dosimetry is essential for stimulating beneficial cellular functions by the NIR laser irradiation system [39,40].

In this study, investigation of time-lapse fluorescence microscopy was first conducted to identify an appropriate irrigation time point for the observation of the cellular response after PBM treatment. Figure 4 showed that the best time point with the most appreciable variation in ΔΨm was within 30 min of PBM treatment period for test fluences (0, 5, and 10 J/cm^2^. This fluence is compatible with result from Chang et al. Rhodamine 123-based observation in auditory cell line, that similar significant increase in mitochondrial membrane potential was induced after PBM treatment (808 nm, 13.5 J/cm^2^) [12]. Thus, the treatment time of 30 min was then adapted for observing the cellular response of hADSC in PBM experiments.

Nevertheless, although there was a tendency that showed a minor increase in the ΔΨm at low fluence (2.5 J/cm^2^) when compared to medium and high fluence (5 and 10 J/cm^2^). However, differences between these two groups were not significant statistically, with *p*-value = 0.78 and 0.74, respectively. Possible reason of the indifference response may due to the fluorescence image sensitivity limit from the ORCA-Flash 4.0 V3 Digital CMOS camera. Thus, to overcome this limitation, an EDCCM camera was used instead, to quantify the PBM effects in ΔΨm and ROS variation for hADSCs. Thereafter, a significant increase in ΔΨ for medium and high fluences (5 and 10 J/cm^2^) was observed, when compared to a fluence of 2.5 J/cm^2^), with corresponding *p*-values at 0.014 and 0.015, respectively. Meanwhile, there was an increase of ROS level peaked at from 5 J/cm^2^, followed by a concomitant decrease at 10 J/cm^2^, contrary to the maintained increase of MMP form 5 J/cm^2^ to 10 J/cm^2^ (see Figure 8 for details). Furthermore, the use of a medium fluence (830 nm, 4.8 W/cm^2^) in PBM was also been observed to induce a significant increase in ATP synthesis in the cerebral cortex, when compared to the region without treatment [41]. Although, these results may point to the possibility that the PBM at lower fluences is capable of inducing pro-surviving expression of redox sensitive transcription factor such as NF-κB [42]. Since the MMP level remained higher level at 10 J/cm^2^ in our result, which was inconsistent with the biphasic pattern observed in cortical neurons under low level laser treatment (810-nm, 0–10 J/cm^2^) [42]. This result may partly explain the intrinsic different in PBM response pattern existed in between cell of various types [43]. However, to elucidate the underlying mechanisms, further confirmatory experiments regarding whether or not at higher fluence, a pro-apoptotic response will be induced or else, are highly suggested.

The assessment of the ΔΨm in living cells has been used to estimate mitochondrial bioenergetic state [30]. Our results (Figure 6) demonstrated that the best performances in the increased ΔΨm were at a fluence greater than 5 J/cm^2^. Indeed, the changes of ΔΨm in the irradiated-hADSCs can also be interpreted through its biological attribute to the absorption of light by internal photoreceptors of the respiratory chain located in the mitochondria [44]. In particular, the ΔΨm is a potential difference generated by proton pumps (Complex I, II, and IV) which plays an imperative role in maintaining the physiological response of the respiratory chain for ATP synthesis [45]. In contrast, due to the mitochondrial permeability transition pore (mPTP) opening, the collapse of the ΔΨm often leads to the releasing of CCO into the cytosol, which causes the mitochondrial membrane depolarization and ATP exhaustion [46,47].

Based on our experiments, a schematic illustration of the mechanisms leading to elevated levels of ROS within irradiated hADSC cells showed a biphasic dosage response. This was indicated earlier as in the case: if insufficient energy is applied, there will be no response (because the minimum threshold has not been met) [43]. On the other hand, if more energy is applied, a threshold will be crossed, and stimulation effects will be achieved; however, if too much energy is applied, the promoting effects will be replaced by inhibition effects [43]. This, in turn, can be beneficial in reducing inflammation, modifying immunological responses.

In addition, although previous studies have proposed a positive efficacy between ΔΨm and ROS production, however, our results in the ΔΨm and ROS (Figure 6 and Figure 7) did not reveal a compatible trend [14,15]. For example, at fluences of 5 and 10 (J/cm^2^), ΔΨm was not shown to be affected and stayed nearly the same, whereas ROS production reached the highest rate at 5 J/cm^2^, then reduced again at 10 J/cm^2^. Such differences might be interpreted via a phenomenon called “proton leak”, which is defined as the dissipation of a protonmotive force (ΔP) in the presence of ATP synthase inhibitor (Oligomycin) [18]. Various effects of “proton leak” were reported in previous studies concerning its regulatory impact on the ΔΨm and ROS. For example, a previous research indicated a decrease in ROS production that induces a proton (H(+)) leak across the mitochondrial inner membrane [48]. On the other hand, others had also demonstrated an increased proton leak that would lead to the raised in ΔΨm exponentially [49]. The results of this study evoke a question of whether ROS and ΔΨm genuinely have a positive correlation or not. In fact, there is a slight decrease in ΔΨm at a fluence of 10 J/cm^2^, but it was not statistically significant in our investigations. A previous study showed a considerable decline in ROS generation was caused by reducing slightly in ΔΨm [50]. Obviously, our results were consistent with the results from previous investigations.

In addition, it is believed that the increased ΔΨm and ROS are correlated to the promotion of vesicle transport [51]. Vesicular transport has been suggested to play an essential role in rebuilding cellular membranes through vesicle-membrane fusion and fission, and carrying proteins to the Golgi apparatus and other molecules from an individual cell to another area [52]. In this study, ROI areas (64 × 64 pixels) were selected to investigate the relationship between mitochondria and vesicle transport (Figure 9). Noted that the number of mitochondria in ROI areas is different between the control group (0 J/cm^2^) and the treatment group (5 J/cm^2^) in Figure 9. Nevertheless, based on using the velocity ratio before and after treatment (V_2_/V_1_) to determine the vesicle transport, the capacity of vesicle transport was consistent among the investigated-hADSCs (Appendix A).

Our results (Figure 10) showed that velocity at 5 J/cm^2^ was observed in the PBM-irradiated hADSCs, implying that changes in ROS and ΔΨm levels could link to the elevation transport activity of intracellular vesicle, as was suggested by a previous study from Zorova et al. [45]. Indeed, most mitochondrial proteins are synthesized in the cytosol and then imported into mitochondria. Therefore, under circumstances that are depressed ΔΨm, protein delivery could be affected due to the limitation in vesicular transport [45]. In addition, since enzymes such as NADH dehydrogenase and cytochrome c oxidase (CCO, Complex IV) had been pointed out to show a corresponding changes in their biochemical properties following PBM [53]. Nitric oxide, which is known to inhibit CCO following two pathways that leads to formation of nitrosyl-derivative (CCO-NO) or nitrite-derivative (CCO-${\mathrm{NO}}_{2}^{-}$) [54]. These derivatives can be further dissociated by photon absorption, a phenomenon known as photodissociation [54]. Therefore, more studies will be needed to further verify ATP and its effects in NO production in future research to study the PBM capacities, especially in the application PBM during diseased or damaged tissues repair. On the other hand, the safe and effective required in stem cell therapy has awakened a new era to develop new cell sorting system to increase the quality of stem cell characteristics before PBM treatment.

Consequently, our studies had demonstrated the effectiveness of PBM in triggering mitochondrial membrane potential and ROS production at the single-cell level. In addition, these single-cell-based research results can further provide detailed response profiles in real-time living cells, such as vesicle transport activities. Significantly, the 5 J/cm^2^ of PBM with a NIR (830 nm) diode laser indicated the best performance in the ΔΨm and ROS generation on human adipose stem cells.

## 5. Conclusions

In summary, our single-cell research demonstrates an increase of the ΔΨm, ROS and vesicle transport at a fluence of 5 J/cm^2^ in the irradiated-hADSCs. These results illustrated the regulation of cellular responses regarding ΔΨm, ROS and vesicle transport on the irradiated-hADSCs, which have potential in cell proliferation, migration, and differentiation in cell-based therapy. We are currently analyzing the PBM effects on hADSC migration at several fluences (2.5, 5.0, and 10 J/cm^2^) using the Culture-Insert 2 Well in µ-Dish 35 mm, and these results will be reported in the near future.

## Figures and Tables

**Figure 1 cells-11-00972-f001:**
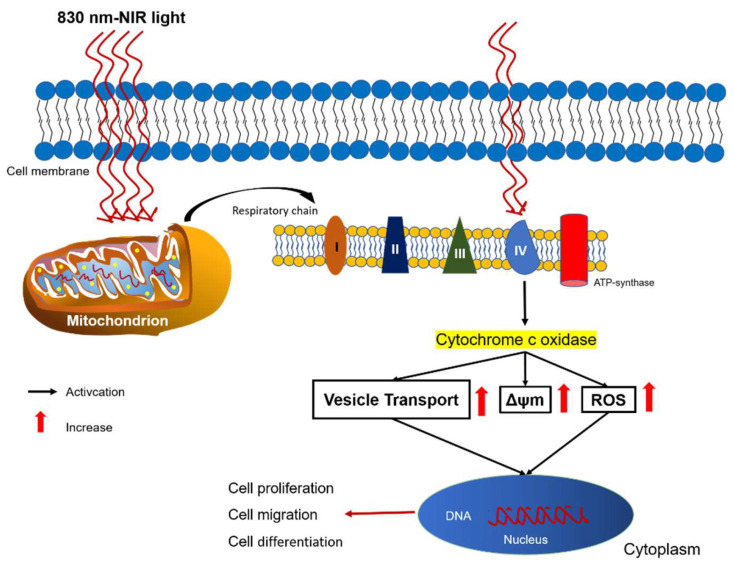
Schematic diagram for mechanisms of photobiomodulation. Cytochrome c oxidase (CCO) is a major photoreceptor, located in the mitochondrial respiratory chain at unit IV in the mitochondria, was released by PBM, followed by a functional change in the original mitochondrial respiratory chain to generate ROS and increased mitochondrial membrane potential (ΔΨm) that plays a crucial role in ATP production through oxidative phosphorylation.

**Figure 2 cells-11-00972-f002:**
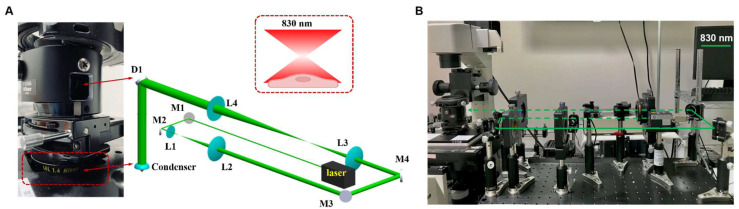
Experimental organization for the single-cell NIR laser irradiation system. (**A**) The proposed experimental setup consists of an 830 nm infrared diode laser, an electrical shutter, laser-focusing optics (condenser), and a specimen holder attaching to the XY-axis motorized stage. The lens pairs, which were used to expand the laser beam eightfold, were combined by two telescopes, including a 1:4 telescope (L1:L2) and 1:2 telescope (L3:L4) with the aim to fill the back aperture of the condenser. The dichroic mirror D1 is located between the visible light source and the condenser to reflect the laser beam into the condenser while allowing visible light to pass for bright field imaging. (**B**) The experimental setup for the proposed single-cell PBM irradiation system.

**Figure 3 cells-11-00972-f003:**
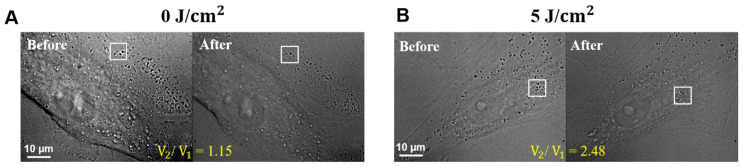
Example for showing the determination and selection of the region of interest (ROI) in the irradiated-hADSCs for vesicle transport analysis. The irradiated-hADSCs (Passage 10) were photographed with 100 of kinetic length and 0.1 s of an exposure time by an EMCCD camera before and after 30 min PBM at two fluences, including (**A**) 0 and (**B**) 5 J/cm^2^.

**Figure 4 cells-11-00972-f004:**
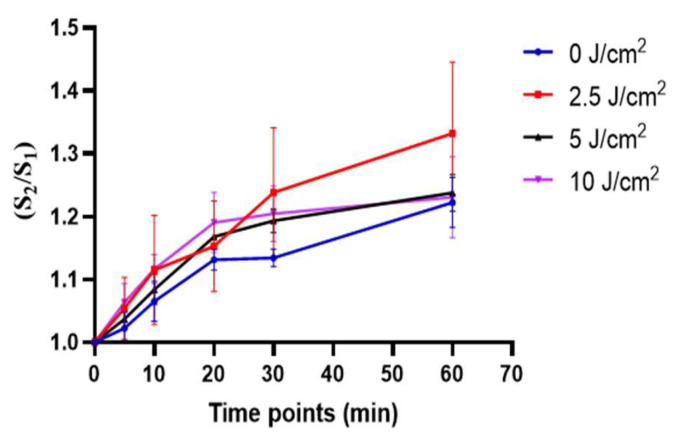
A plot of time-lapse Rh123 of mitochondrial membrane potential in PBM. Values demonstrated the ratio of mean intensity before and after 0, 10, 20, 30, 60 min PBM (*n* = 3). Statistical analysis (one-way ANOVA) for significant differences between after 30 PBM and other time points observed at all fluences, including 0, 2.5, 5, and 10 J/cm^2^ (*p* ≤ 0.05).

**Figure 5 cells-11-00972-f005:**
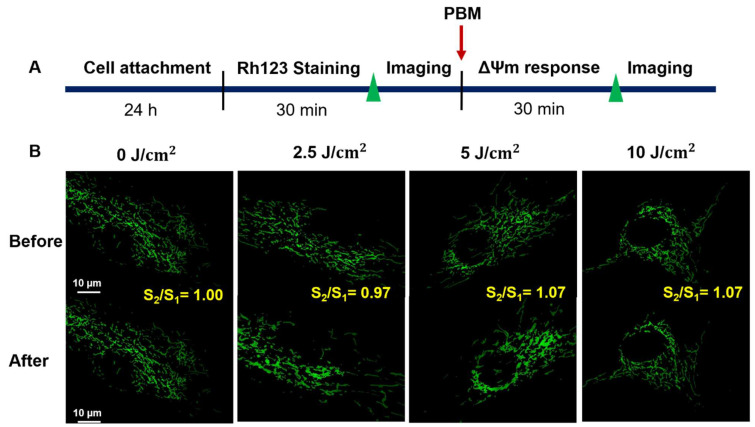
(**A**) Experimental procedures in investigating the ΔΨm in the irradiated-hADSCs (**B**) Fluorescence microscopy images of the promotion of the mitochondrial membrane potential in hADSCs (Passage 10) after 30 min PBM at various fluences, including 0, 2.5, 5, and 10 J/cm^2^. These cells were automatically selected and captured after 30 min by EMCCD camera.

**Figure 6 cells-11-00972-f006:**
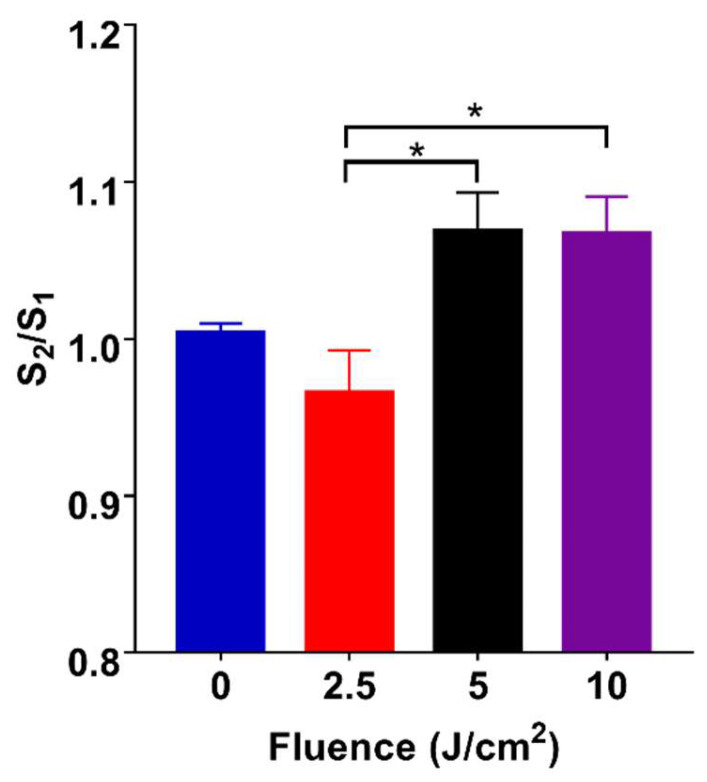
Mitochondrial membrane potential in hADSCs (Passage 10) after 30 min PBM at various fluences, including 0, 2.5, 5, and 10 (J/cm^2^). Values represented as the ratio of mean fluorescence intensity before and after 30 min PBM (S_2_/S_1_ ± SEM, *n* = 8). Statistical analysis (one-way ANOVA) for significant differences between 2.5 J/cm^2^) and two fluences of 5 and 10 J/cm^2^ (* *p* ≤ 0.05).

**Figure 7 cells-11-00972-f007:**
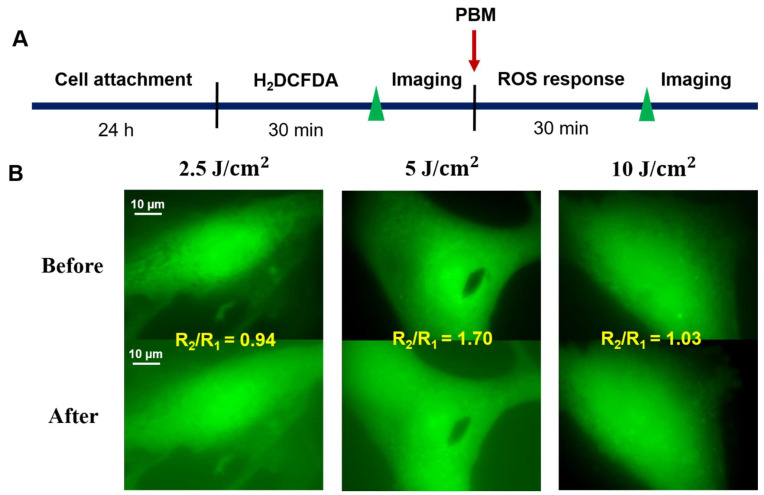
(**A**) Experimental procedures in investigating the ROS production in the irradiated-hADSCs (Passage 10) (**B**) ROS marker fluorescence microscopy pictures were acquired in the hADSCs after 30 min. These cells were spontaneously picked and captured by an EMCCD camera following various time points of PBM at fluences of 2.5, 5, and 10 J/cm^2^. The scale bar denotes a distance of 10 µm.

**Figure 8 cells-11-00972-f008:**
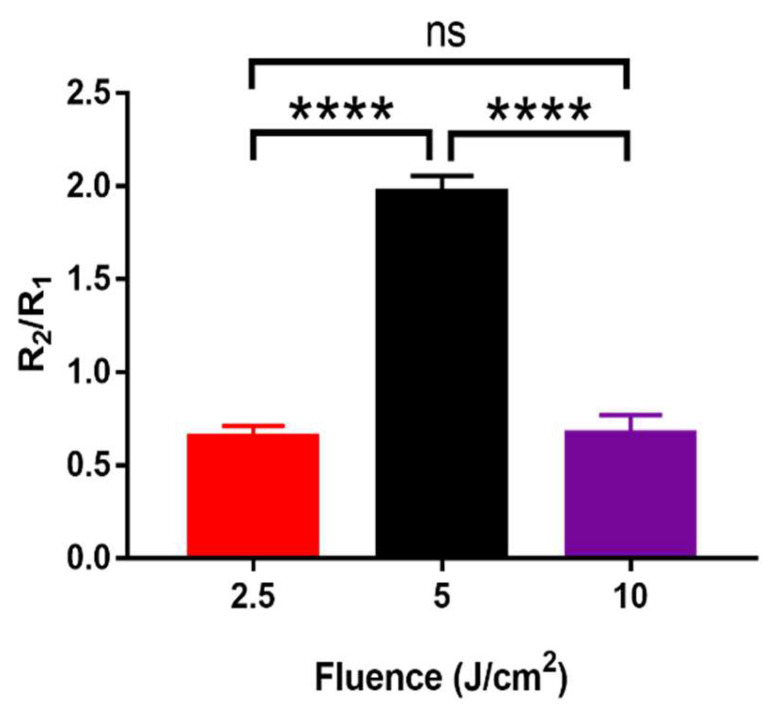
Bar charts representing the effects of PBM on intracellular ROS acquired in the hADSCs after 30 min PBM. These cells were spontaneously picked and photographed by an EMCCD camera following PBM at fluences of 2.5 J/cm^2^ (Red), 5 J/cm^2^ (Black), and 10 J/cm^2^ (Purple). The values are the mean intensity ratio (mean ± SEM, *n* = 6). Statistical analysis (one-way ANOVA) showed significant changes in this study (**** *p* ≤ 0.0001; ns: *p* > 0.05).

**Figure 9 cells-11-00972-f009:**
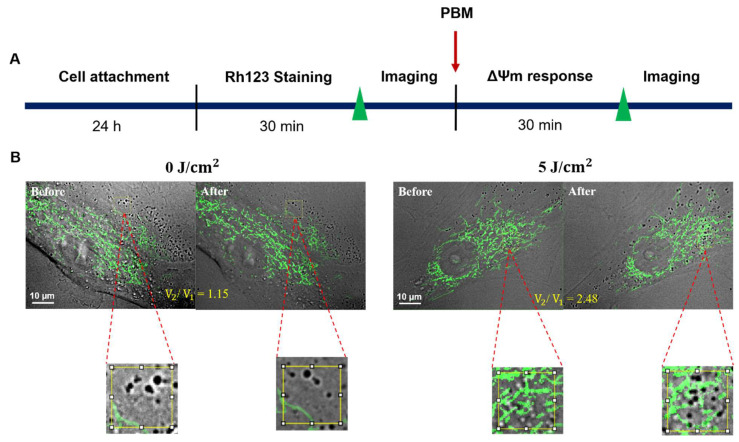
(**A**) Experimental procedures in vesicle transport investigation on the irradiated-hADSCs (**B**) The mergence between bright field and fluorescence images in promoting the vesicle transport in the irradiated-hADSCs (Passage 10) after 30 min treatment at a fluence of 5 (J/cm^2^) compared to control 0 (J/cm^2^). The ROI area (64 *×* 64 pixels) was identified by ImageJ, followed by cutting it and measure the velocity using (Matlab R2021a).

**Figure 10 cells-11-00972-f010:**
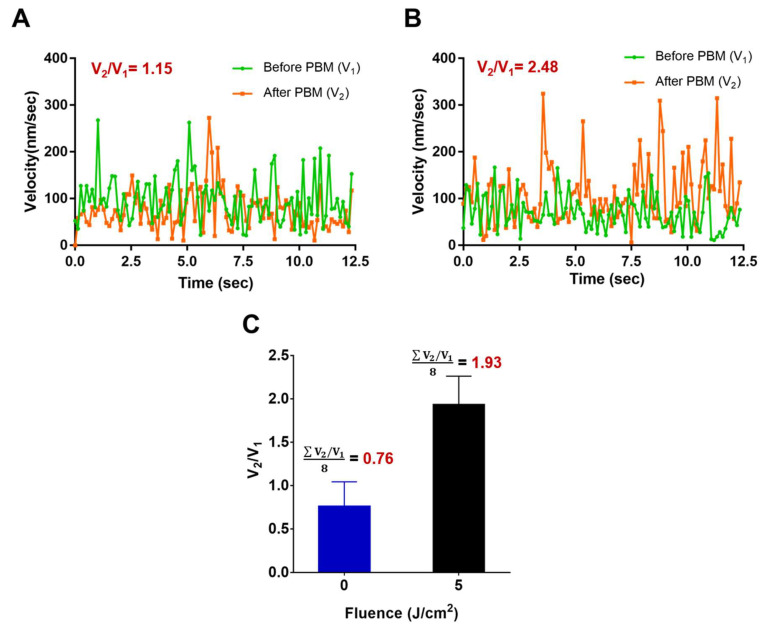
Effects of PBM in vesicle transport on hADSCs. Values were expressed as the velocity’s ratio of vesicle transport before (V_1_) and after PBM (V_2_) according to time (V_2_/V_1_ ± SEM). (**A**) The velocity of the control group (0 J/cm^2^); (**B**) treatment group (5 J/cm^2^) at single-cell level. (**C**) Vesicle transport in the irradiated-hADSCs (*n* = 8) after 30 min PBM between control group of 0 J/cm^2^ and PBM group of 5 J/cm^2^. The ensemble average of the velocity ratio (V_2_/V_1_) for the control and PBM groups were 0.76 and 1.93, respectively (*p*-value = 0.0117).

## Data Availability

Raw image data and analyzed data can be found in the attached folder and these data were generated during the study.

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
