# Peer review of "Single Cell Effects of Photobiomodulation on Mitochondrial Membrane Potential and Reactive Oxygen Species Production in Human Adipose Mesenchymal Stem Cells"

_cells, 2022, doi:10.3390/cells11060972_

Round 1

Reviewer 1 Report

In this work, the authors proposed a single-cell near-infrared laser irradiation system to demonstrate biological effects, including alteration of the ΔΨm, ROS and, and vesicle transport on hADSCs. The experiments are comprehensive and have been performed. The obtained results have good support for the findings of the study.

Author Response

General Comment: This manuscript illustrated the biological efficacies in changing ΔΨm, ROS, and vesicle transport on hADSCs after PBM using the single-cell NIR laser irradiation system. In addition, the reported results were valuable for supporting this research.

General Response: The authors would like to thank Reviewer #1 for providing many positive comments for our manuscript. These positive comments give us a strong spirit and motivation for our dedication and efforts in this study. Furthermore, we believe that this study is a good beginning to continue pursuing the following promising studies in the near future.

Reviewer 2 Report

The present study investigated that a specific range of PBM can regulate mitochondrial membrane potential (ΔΨm), reactive oxygen species (ROS), and vesicle transport in living human adipose mesenchymal stem cells (hADSCs). And the authors insist that PBM has an efficacy and use in cell therapy by its positive biological effects.

Although the authors demonstrated that PBM increases ΔΨm, ROS, and vesicle transport in ASCs using several methods, they are not enough to support the conclusion of this article; “efficacy and use of PBM in stimulating the positive 41 biological effects, which can promote cell proliferation, migration, and differentiation in 42 cell therapy”. Based on these experimental demonstrations here, the readers cannot understand whether increased ΔΨm, ROS, and vesicle transport by PBM in this article can affect positive biological effects or not. Therefore, there is a lack of evidence to support the hypotheses insisted by the authors generally.

- Most descriptions in supplementary figures do not have enough information.

Author Response

General Comment: Though the reported results demonstrated the regulation in mitochondrial membrane potential (ΔΨm), reactive oxygen species (ROS), and vesicle transport in the irradiated- hADSCs, there is a lack of clear evidence to prove these changes related to the PBM-based efficacies on positive biological effects, including cell proliferation, migration and differentiation in cell therapy, which results in being confusing readers. In addition, the insufficient detailed information in supplement figures was pointed down.

General Response:

The authors would like to thank Reviewer #2 for the helpful comments that allowed us to revise the manuscript. We have made several revisions to the manuscript following Reviewer #2’s suggestions, and your concern has been carefully addressed, as can be seen below. Modifications are the yellow highlight or red color words in the markedly revised manuscript to help the Reviewer and the Editor more straightforwardly check on the changes we made.

------------------------------------------------------------------------------------------------------------------------

Comments from Reviewer #2:

Major Concerns

  • Although the authors demonstrated that PBM increases ΔΨm, ROS, and vesicle transport in ASCs using several methods, they are not enough to support the conclusion of this article; “efficacy and use of PBM in stimulating the positive biological effects, which can promote cell proliferation, migration, and differentiation in cell therapy”. Based on these experimental demonstrations here, the readers cannot understand whether increased ΔΨm, ROS, and vesicle transport by PBM in this article can affect positive biological effects or not. Therefore, there is a lack of evidence to support the hypotheses insisted by the authors generally.

Author’s Response: Thanks for your insightful comments on our manuscript.

  We agree with this comment. The present study proves that a fluence of 5 J/cm2 of PBM significantly enhanced the ΔΨm, ROS, and vesicle transport phenomena, where the increase of ΔΨm, ROS, and vesicle transport is one of several elements leading to cell proliferation, migration, and differentiation. For example, according to the research of Martínez-Reyes et al., ROS production is necessary for cell proliferation, plus the relationship between ΔΨm and ROS mentioned in their study [1]. In addition, the previous study also proves the relationship between the increase of ΔΨm, ROS, and cell proliferation/migration [2]. However, our present study does not have results related to the investigation on cell proliferation, migration, and differentiation. We are currently applying the fluence of 5 J/cm2 of PBM to prove whether the increase of ΔΨm, ROS, and vesicle transport is related to cell proliferation, migration, and differentiation. Therefore, to avoid confusion regarding the efficacy of PBM in regulating cellular response (ΔΨm, ROS, and vesicle transport) in hADSCs, we have re-written some sentences in the markedly revised manuscript, which are also listed below.

  • Line 40-42: “These findings demonstrate the efficacy and use of PBM in stimulating the positive biological effects, which can promote cell proliferation, migration, and differentiation in cell therapy.”

Author’s Modification: These findings demonstrate the efficacy and use of PBM in regulating ΔΨm, ROS, and vesicle transport, which have potential in cell proliferation, migration, and differentiation in cell-based therapy.

  • Line 44: “cell therapy.”

Author’s Modification: cell therapy

  • Line 102-104: “From that, one of the best fluence from 2.5 to 10 J/cm2 can be identified for the ΔΨm and ROS promotion, which eventually stem cell proliferation, migration, and differentiation.”

Author’s Modification: From that, one of the best fluence from 2.5 to 10 J/cm2 can be identified for the ΔΨm and ROS promotion, which might link to the potential cellular functions, including stem cell proliferation, migration, and differentiation.

  • Line 595-599: “These results indicated the enormous potential of the use PBM in stimulating the positive biological effects, which promotes cell proliferation, migration, and differentiation in cell-based therapy.”

Author’s Modification: These results illustrated the regulation of cellular responses regarding ΔΨm, ROS and vesicle transport on the irradiated-hADSCs, which have potential in cell proliferation, migration, and differentiation in cell-based therapy.

Reference

[1] Martínez-Reyes, I., et al., TCA cycle and mitochondrial membrane potential are necessary for diverse biological functions. Molecular cell, 2016. 61(2): p. 199-209.

[2] Zorov, D.B., M. Juhaszova, and S.J. Sollott, Mitochondrial reactive oxygen species (ROS) and ROS-induced ROS release. 578 Physiological reviews, 2014. 94(3): p. 909-950.

  • The insufficient detailed information in supplement figures was pointed down.

Author’s Response: Thank you for pointing out a lack of detailed information in our supplement data. It is essential to describe what these figures mean, which helps readers comprehend the clues of our manuscript and how the reported results come up. Thus, the authors added more descriptions for some of the figures in supplement data, which are listed in the Supplementary Material and are also listed below.

Author’s Modification:

  • Fig S1 + S2:

By investigating the range of the Rh123’s concentration, the best concentration for Rh123’s performance can be determined and used for further experiments. In addition, the effect of Rh123’s concentration has a significant impact on mitochondrial staining, which leads to whether the mitochondrial membrane is performed clearly via Rh123 staining or not. Most importantly, the resolution of images influences measuring the signal intensity of mitochondrial membrane potential. Thus, determining the proper dye concentration is necessary for profound studies. In this study, Rh123 dye was investigated at various concentrations 1.0, 5.0, and 10 µM, followed by automatically selecting and capturing five single cells at each concentration using an ORCA-Flash 4.0 V3 Digital CMOS camera. These images were then done image processing for subtracting background noise and keeping the mitochondrial structure by MATLAB software (Fig S1). The results demonstrated that the mean dimensionless ratio at the Rh123’s concentration of 1.0, 5.0, 10 µM were 4.02; 6.44 and 3.44 in turns. The Rh123 concentration of 5 µM showed the best performance of Rh123 dye in the hADSC’s mitochondrial staining (Fig. S2) 

  • Fig S3: By setting up of a treatment zone that refers to the hADSCs size, the pulsed laser beams’ spot area was created, and its area was then calculated by ImageJ software, 3 x 10-6 (cm2) in this case (Fig.S3). From that, a good range of fluences (2.5; 5; 10 J/cm2) in PBM was then identified by changing different treatment times, including 10, 20, 40 seconds and keeping input power (20 mW) and area (3 x 10-6 µm2) stable. In addition, a pass filter (OD = 4) was used to reduce the energy of a laser beam of wavelength 830 nm.
  • Fig S4: A wide range of fluency, which is used in photobiomodulation experiments, is calculated based on three main elements, including power (W), time (s), and area (cm2). A formula (1) is shown above:

 (1)

For one, power (J), which refers to real power in the chamber, is quantified based on power input and transmission of Nikon microscope objectives, which is investigated and indicated only 59% transmission capacity of wavelength 830 nm in the previous studies. The results are calculated according to the formula (Power in chamber = power output x 100/59). An wide investigated range of power input is 10, 20, 30, 40, 50, 60, 70, 80, 90, 100 (mW) and the real power in chamber is 3.64; 7.36; 11.73; 15.9; 19.95; 24.03; 28.12; 32.2;36.12; 40.25 (mW), respectively. The diagram illustrates that a straight line (y = 0.4073x – 0.4433) means a continuous change in the values at different power input values, and an R-squared value equal to one indicated a slight significance in analysis processing (Fig.S4).

  • Fig S5: To enhance the precise calculation in mitochondrial membrane potential (ΔΨm), background noise elimination is necessary before measuring the mean fluorescence intensity of the Rh123 before and after PBM. In addition, the original mitochondrial structure was observed clearly without image issues by developing a code and inputting it into Matlab to remove background noise. Note that an appropriate radius is a primary key for eliminating background noise in these cases. Taking Fig S5 as an example, the radius of 20 is unsuitable due to the background noise that existed and was observed in the Mitochondrion image. In contrast, the mitochondrial structure was observed clearly with a radius of 10.

Reviewer 3 Report

Li-Chern Pan, Nguyen Le Thanh Hang and colleagues present the manuscript "Single Cell Effects of Photobiomodulation on Mitochondrial Membrane Potential and Reactive Oxygen Species Production  in Human Adipose Mesenchymal Stem Cells"

This is an interesting premise for a manuscript with the application of PBM to hADSCs. However, with many manuscripts that are submitted a range of revisions are needed to bring the paper to a publication level. This manuscript does require some work before it is publishable. There are ranges for minor to major corrections and clarifications needed.

  1. In the abstract the statement “These findings demonstrate the efficacy and use of PBM in stimulating the positive biological effects, which can promote cell proliferation, migration, and differentiation in cell therapy”, is speculative at best as there is no evidence in this manuscript that proliferation, migration and differentiation is affected.

  1. There is an important query that pertains to the media composition, that is not a standard or established formulation. The use of MilliporeSigma’s mesenchymal-LS expansion medium which main feature is low serum at 2%. However, the authors used an additional FBS to 10%. Compounding this is the use of rhFGF-b, Ascorbic acid, Hydrocortisone Hemisuccinate and Insulin from the supplement. Was this an erroneous typographical error? Also, it was referenced from 3 sources below, all of which do not use this media composition. All utilise DMEM with glutamine and 10% FBS.

Helmy, M.A., et al., A protocol for primary isolation and culture of adipose-derived stem cells and their phenotypic profile. Alexandria 506 Journal of Medicine, 2020. 56(1): p. 42-50. 507, Wang, J.M., et al., Isolation, culture and identification of human adipose-derived stem cells. Experimental and therapeutic medicine, 508 2017. 13(3): p. 1039-1043. 509 and Mahmoudifar, N. and P.M. Doran, Mesenchymal stem cells derived from human adipose tissue, in Cartilage tissue engineering. 2015, 510 Springer. p. 53-64.

  1. There is a stark lack of any qualitative or quantitative biological assays

There is a litany of typographical errors with incomplete words or incorrect adjective usage that are too numerous to list here. However, the biggest areas of notice are results and discussion as well as figure legends. Some examples below

Line 267, Sentence requires revision “Perhap, because of different kinds of camera to capture fluorescence images in these cases.”

Figure 3. “Example for showhing the determination and selection of the region of interest (ROI) in”

  1. The conclusions final sentence is not supported by the findings of this paper,

While an interesting application, this manuscript is not suitable for publication in this Journal.

Author Response

General Comment: Though this manuscript evokes the potential of PBM application in hADSCs, it is required to revise and modify some mistakes carefully before publication. Particularly, though the reported results demonstrated the regulation in mitochondrial membrane potential (ΔΨm), reactive oxygen species (ROS), and vesicle transport in the irradiated- hADSCs, there is a lack of clear evidence to prove these changes related to the PBM-based efficacies on positive biological effects, including cell proliferation, migration and differentiation in cell therapy, which results in being confusing readers. In addition, a wide range of minor to major corrections, which need to change, is listed particularly below.

General Response: The authors would like to thank Reviewer #3 for your helpful comments that allowed us to revise our manuscript. We have made several revisions to the manuscript following your suggestions, and your concern has been carefully addressed, as can be seen below. Modifications are the yellow highlight or red color words in the markedly revised manuscript to help the Reviewer and the Editor more straightforwardly check on the changes we made.

------------------------------------------------------------------------------------------------------------------------

Comments from Reviewer #3:

Major Concerns

  • In the abstract the statement “These findings demonstrate the efficacy and use of PBM in stimulating the positive biological effects, which can promote cell proliferation, migration, and differentiation in cell therapy”, is speculative at best as there is no evidence in this manuscript that proliferation, migration and differentiation is affected.

Author’s Response: Thanks for your insightful comments on our manuscript.

  We agree with this comment. The present study proves that a fluence of 5 J/cm2 of PBM significantly enhanced the ΔΨm, ROS, and vesicle transport phenomena, where the increase of ΔΨm, ROS, and vesicle transport is one of several elements leading to cell proliferation, migration, and differentiation. For example, according to the research of Martínez-Reyes et al., ROS production is necessary for cell proliferation, plus the relationship between ΔΨm and ROS mentioned in their study [1]. In addition, the previous study also proves the relationship between the increase of ΔΨm, ROS, and cell proliferation/migration [2]. However, our present study does not have results related to the investigation on cell proliferation, migration, and differentiation. We are currently applying the fluence of 5 J/cm2 of PBM to prove whether the increase of ΔΨm, ROS, and vesicle transport is related to cell proliferation, migration, and differentiation. Therefore, to avoid confusion regarding the efficacy of PBM in regulating cellular response (ΔΨm, ROS, and vesicle transport) in hADSCs, we have re-written some sentences in the markedly revised manuscript, which are also listed below.

  • Line 40-42: “These findings demonstrate the efficacy and use of PBM in stimulating the positive biological effects, which can promote cell proliferation, migration, and differentiation in cell therapy.”

Author’s Modification: These findings demonstrate the efficacy and use of PBM in regulating ΔΨm, ROS, and vesicle transport, which have potential in cell proliferation, migration, and differentiation in cell-based therapy.

  • Line 44: “cell therapy.”

Author’s Modification: cell therapy

  • Line 102-104: “From that, one of the best fluence from 2.5 to 10 J/cm2 can be identified for the ΔΨm and ROS promotion, which eventually stem cell proliferation, migration, and differentiation.”

Author’s Modification: From that, one of the best fluence from 2.5 to 10 J/cm2 can be identified for the ΔΨm and ROS promotion, which might link to the potential cellular functions, including stem cell proliferation, migration, and differentiation.

Reference

[1] Martínez-Reyes, I., et al., TCA cycle and mitochondrial membrane potential are necessary for diverse biological functions. Molecular cell, 2016. 61(2): p. 199-209.

[2] Zorov, D.B., M. Juhaszova, and S.J. Sollott, Mitochondrial reactive oxygen species (ROS) and ROS-induced ROS release. 578 Physiological reviews, 2014. 94(3): p. 909-950.

  • There is an important query that pertains to the media composition, that is not a standard or established formulation. The use of MilliporeSigma’s mesenchymal-LS expansion medium which main feature is low serum at 2%. However, the authors used an additional FBS to 10%. Compounding this is the use of rhFGF-b, Ascorbic acid, Hydrocortisone Hemisuccinate and Insulin from the supplement. Was this an erroneous typographical error? Also, it was referenced from 3 sources below, all of which do not use this media composition. All utilise DMEM with glutamine and 10% FBS.

      Author’s Response: Thanks for your insightful comments on our manuscript.

  Firstly, the authors are sorry about making a mistake in writing regarding we wrote wrongly the origin of the cell line we used for our studies. In particular, we used the human mesenchymal adipose stem cell line (HMSC-ad 7510) provided by ScienCell Research Laboratories (USA) instead of using the human mesenchymal adipose stem cell line (SCC038) provided by EMD Millipore (USA).

  Honestly, we first bought SCC038 and all products for cell culture from EMD Millipore (USA), then followed the protocol of cell culture (2% FBS) to conduct cell culture. Nevertheless, the cells grew slowly from early passages (Passage 3); it takes nine days to reach 80% confluency after thawing the cells from the vial (Fig. 1). In addition, the morphology of hADSCs (passages 5 & 6) changed, plus the proliferation speed was slow; eventually, they died (Fig 2, 3; respectively).

Fig 1. Human Adipose Mesenchymal Stem Cells (Catalog No. SCC038). The hADSCs (Passage 3), which reached 80% confluent after nine days incubation, were plated at 5000 cells/cm2.

Fig 2. Human Adipose Mesenchymal Stem Cells (Catalog No. SCC038). The hADSCs (Passage 5) were plated at 5000 cells/cm2. As our observation, the hADSCs changed their morphology, grew slowly, and followed by being died steadily at the ninth-day incubation, though the fresh medium was changed every 3-4 days. Then, the subculturing process was conducted after nine days of incubation.

Fig 3. Human Adipose Mesenchymal Stem Cells (Catalog No. SCC038). The hADSCs (Passage 6) were plated at 5000 cells/cm2. As our observation, the proliferation process of hADSCs (Passage 6) was the same as the hADSCs (Passage 5), but they died after nine days of incubation.

  Therefore, we tried to culture the same cell line (hADSCs), but different the origin of the cell line at that time. Mainly, we used the human mesenchymal adipose stem cell line (HMSC-ad 7510) provided by ScienCell Research Laboratories (USA) to conduct further experiments. In the beginning, we still used materials provided EMD Millipore (USA), including human mesenchymal-LS expansion medium (MilliporeSigma, Burlington, USA) containing 5 ng/mL rhFGF-b, 50 µg/mL Ascorbic acid, 7.5 nM L-glutamine, 1.0 µg/mL Hydrocortisone Hemisuccinate, 5 µg/mL rh Insulin and 2% fetal bovine serum (FBS) in the beginning (Fig. 4A). Moreover, due to the low proliferation rate of hADSCs, we try to increase 2% FBS into 10% FBS to see whether there is any difference in cell growth or not. (Fig. 4B). We observed the hADSCs grew more stable; thus, 10% FBS had been applied for cell culture from that time. Our mistake is the lack of results for investigating the effects of FBS concentration for regulating hADSC culture in this case. Though all figures might not convince reviewers that the impact of using 10% FBS was better than that of 2% FBS in this case, we tried our best to provide all of what results in we had to tell reviewers that we tried several ways to optimize our protocol.

Fig 4. Human Adipose Mesenchymal Stem Cells (HMSC-ad 7510 line). (A) the hADSCs (Passage 9) were cultured using 2% FBS after five days incubation. (B) the hADSCs (Passage 15) were cultured using 10% FBS after one days incubation.

In general, the modification of a procol should investigated

  • Line 170-172: “Human mesenchymal adipose stem cell line used for this study was provided by EMD Millipore (USA) and cryopreserved as secondary cells from a single donor of human adipose tissue.”

Author’s Modification: Human mesenchymal adipose stem cell line (HMSC-ad 7510) used for this study was provided by ScienCell Research Laboratories (USA).

  • There is a stark lack of any qualitative or quantitative biological assays. There is a litany of typographical errors with incomplete words or incorrect adjective usage that are too numerous to list here. However, the biggest areas of notice are results and discussion as well as figure legends.
  • Line 441-584: “Though PBM has widely emerged in cell-based therapy in recent times….. cell migration, and differentiation on human adipose stem cells in stem cell therapy”

Author’s Response: Thanks for your insightful comments on our manuscript. The authors have modified the discussion part. Please see the markedly revised manuscript.

  • The conclusions final sentence is not supported by the findings of this paper, While an interesting application, this manuscript is not suitable for publication in this Journal.
  • Line 595-599: “These results indicated the enormous potential of the use PBM in stimulating the positive biological effects, which promotes cell proliferation, migration, and differentiation in cell-based therapy.”

Author’s Modification: These results illustrated the regulation of cellular responses regarding ΔΨm, ROS and vesicle transport on the irradiated-hADSCs, which have potential in cell proliferation, migration, and differentiation in cell-based therapy.

Reviewer 4 Report

This is an interesting study. My major concerns are related with the statistics. The authors overuse paired t-test for comparisons, leading to significant differences that, in my opinion do not exist. In my opinion, the conclusions about Figure 1 need to be validated using two-way ANOVA. Figure 4 using one-way ANOVA and so on. In Figure 8, R2/R1 as a function of fluence, it is clear that 5 J/cm2 values are different from the other 2 values, but the statistics is not correct.

Statistics must be corrected to support several conclusions.  

Author Response

General Comment: Overall, this manuscript is exciting research. Nevertheless, the statistical analysis should be verified and corrected for specific experiments to support various conclusions. In addition, whether or not there are some copyright issues regarding permission for using figures 1 and 5 in this manuscript.

General Response: The authors would like to thank Reviewer #4 for your helpful comments that allowed us to revise our manuscript. We have made several revisions to the manuscript following your suggestions, and your concern has been carefully addressed, as can be seen below. Modifications are the yellow highlight or red color words in the markedly revised manuscript to help the Reviewer and the Editor more straightforwardly check on the changes we made.

------------------------------------------------------------------------------------------------------------------------

Comments from Reviewer #4:

Major Concerns

  • This is an interesting study. My major concerns are related with the statistics. The authors overuse paired t-test for comparisons, leading to significant differences that, in my opinion do not exist. In my opinion, the conclusions about Figure 1 need to be validated using two-way ANOVA. Figure 4 using one-way ANOVA and so on. In Figure 8, R2/R1 as a function of fluence, it is clear that 5 J/cm2 values are different from the other 2 values, but the statistics is not correct. Statistics must be corrected to support several conclusions.

Author’s Response: Thanks for your insightful comments on our manuscript.

The authors agreed with your comments. We re-do the statistics for the results from figure 4, 6 and 8. Modifications were conducted following the reviewer’s suggestions and listed in detail below:

  • Line 102-104: “From that, one of the best fluence from 2.5 to 10 J/cm2 can be identified for the ΔΨm and ROS promotion, which eventually stem cell proliferation, migration, and differentiation”

Author’s Modification: From that, one of the best fluence from 2.5 to 10 J/cm2 can be identified for the ΔΨm and ROS promotion, which which might link to the potential cellular functions, including stem cell proliferation, migration, and differentiation [17].

  • Line 106-125: Figure 1

Author’s Modification:

  • Line 131-132: “From that, it can promote cellular responses, including cell proliferation, 131 migration and adhesion [17]”

Author’s Modification: From that, it can promote cellular responses, including cell proliferation, 131 migration and adhesion [17]

  • Line 264-272: “All data were analyzed and expressed as the mean ± sem of the mean for various independent experiments using GraphPad Prism 8.3.0. The comparison between the control and treatment PBM groups were statistically measured using the paired t-test. Differences were significant at p ≤ 0.05. All data are presented as the mean ± standard error of the mean (SEM).”

Author’s Modification: All data were analyzed and expressed as the mean ± standard error of the mean (SEM) for various independent experiments using GraphPad Prism 8.3.0. One-way analysis of variance (ANOVA) with Tukey HSD (honestly significant difference) post-hoc test was used to assess differences among the groups; p < 0.05 and p < 0.01 were considered statistically significant and statistically highly significant, respectively. In addition, the effects of PBM on vesicle transport was evaluated using the paired t-test.

  • Line 279-290: “The results showed that there was significant difference of ΔΨm between PBM groups and control groups (p ≤ 0.05), whereas the ΔΨm was not statistically changed among treatment groups in this study (p > 0.05) (Figure. 4). A noted point is that the rate of ΔΨm has a steady and linear rise within first 20 minutes of PBM treatment at 2.5; 5 and 10 (J/cm2) and then increases with slower rate after 20 minutes, except for the one with dose of 2.5 (J/cm2).”

Author’s Modification: An appropriate time point was identified by comparing the change in ΔΨm at a wide range of different timepoint. Figure 4 showed that the increased ΔΨm at a time point of 20 mins had a significant difference compared to 0 mins at the fluences of 0 and 5 (J/cm2) (p < 0.05), while it was not at other fluences (p > 0.05). For a fluence of 10 J/cm2, the rate of ΔΨm was increased statistically at a time point of 30 and 60 mins PBM. In addition, there was no significant difference in ΔΨm between 30 and 60 mins PBM at all groups (p > 0.05). In addition, there was no considerable difference between control group and treatment groups at time point of 30 mins PBM (p > 0.05) on the irradiated-hADSCs.

  • Line 308-311: “Statistical analysis (Pair t-test) for significant differences between control group and 2.5, 5, and 10 J/cm2 treatment groups (p ≤ 0.05), while there are no significant among treatment groups (p > 0.05)”

Author’s Modification: Statistical analysis (one-way ANOVA) for significant differences between a time point of 30 mins PBM and other time points observed at all fluences, including 0, 2.5, 5, and 10 J/cm2 (p ≤ 0.05).

  • Line 318-324: “In Figure 6, these results demonstrated that PBM stimulated the statistical change in ΔΨm at fluences of 5 and 10 (J/cm2) with the same ratio of S2/S1 = 1.07 compared to control group S2/S1 = 1.00 (p < 0.05). In constrast, the ΔΨm of 2.5 J/cm2 was not, S2/S1 = 0.97 (p > 0.05). Obviously, the level of laser energy impacts on biological responses after irradiation [31]. Besides, there was statistically no significant difference between the fluence of 5 and 10 (J/cm2), and similarly between low fluence (2.5 J/cm2) and control group (0 J/cm2) in ΔΨm.”

Author’s Modification: In figure 6, these results demonstrated that PBM stimulated the statistical change in ΔΨm at fluences of 5 and 10 (J/cm2) with the same ratio of S2/S1 = 1.07 compared to 2.5 J/cm2 S2/S1 = 1.00 (p < 0.05). Besides, there was statistically no difference between the fluence of 5 and 10 (J/cm2), and similarly between control group (0 J/cm2) and treatment groups in ΔΨm.

  • Line 330-331: “The above results showed a significant difference between 2.5 J/cm2 and control group (0 J/cm2).”

Author’s Modification: The above results showed there was no considerable difference among treatment groups at time point of 30 mins PBM (p > 0.05) on the irradiated-hADSCs.

  • Line 357-365: Figure 6

Author’s Modification:

  • Line 368-371: “Statistical analysis (Pair t-test) for significant differences between control group (0 J/cm2) and treatment groups of 5 and 10 J/cm2 (p ≤ 0.05), whereas 2.5 J/cm2 was not (P > 0.05).”

Author’s Modification: Statistical analysis (one-way ANOVA) for significant differences between 2.5 J/cm2) and two fluences of 5 and 10 J/cm2 (*, p ≤ 0.05).

  • Line 381-384: “Pair t-test analysis was performed that showed a significant difference between 2.5 and 5 J/cm2 and 5 and 10 J/cm2. There was no significant difference on 2.5 and 10 J/cm2

Author’s Modification: One-way ANOVA analysis was performed in this study. There were highly significant differences between 5 J/cm2 compared to two fluences of 2.5 and 10 (J/cm2), p ≤ 0.0001, while there was no significant difference between 2.5 and 10 J/cm2.

  • Line 395-402: Figure 8

Author’s Modification:

  • Line 406-408: “Statistical analysis (Pair t-test) showed significant changes between treatment groups (p ≤ 0.05).”

Author’s Modification: Statistical analysis (one-way ANOVA) showed significant changes in this study (****, p ≤ 0.0001; ns, p > 0.05).

  • Line 457-460: “The results (Figure 4) showed that the best performances in ΔΨm was within 30 mins PBM for all the investigated fluences, followed by the unchanged ΔΨm after 30 mins PBM”

Author’s Modification: Figure 4 showed that the best time point with the most appreciable variation in ΔΨm was within 30 minutes of the PBM treatment period for test fluences (0, 5, and 10 J/cm2).

  • Line 467-470: “There is a clear and prominent variation in the ΔΨm at low fluence (2.5 J/cm2) when compared to medium and high fluences (5 and 10 J/cm2), but these differences were not significant, p-value = 0.13 and 0.5, respectively.”

Author’s Modification: Nevertheless, although there was a tendency that showed a minor increase in the ΔΨm at low fluence (2.5 J/cm2) when compared to medium and high fluence (5 and 10 J/cm2). However, differences between these two groups were not significant statistically, with p-value = 0.78 and 0.74, respectively.

  • Line 475-478: “In addition, our results further illustrated the significant promotion in ΔΨ at treatment groups 2.5, 5 and 10 J/cm2, when compared to control (0 J/cm2), p-value, 0.03 0.04, 0.02, respectively.”

Author’s Modification: Thereafter, a significant increase in ΔΨ for medium and high fluences (5 and 10 J/cm2) was observed, when compared to a fluence of 2.5 J/cm2, with corresponding p-values at 0.014 and 0.015, respectively

  • Line 595-599: “These results indicated the enormous potential of the use PBM in stimulating the positive biological effects, which promotes cell proliferation, migration, and differentiation in cell-based therapy.”

Author’s Modification: These results illustrated the regulation of cellular responses regarding ΔΨm, ROS and vesicle transport on the irradiated-hADSCs, which have potential in cell proliferation, migration, and differentiation in cell-based therapy.

Round 2

Reviewer 2 Report

Although the authors described more information and corrected sentences, it seems that there are no direct evidences to support the potential positive effects of PBM. As I understand correctly, the supporting description the authors provided is from other cell types and PBM range. And if this article contains only information on the changes of ΔΨm, ROS, and vesicle transport in ASCs, i am not sure that these information can meet the significance of this journal "cells". 

Author Response

The authors would like to thank Reviewer #2 for the helpful comments that allowed us to conduct experiments to provide direct evidence to support the potential positive effects of PBM. We are currently analyzing the PBM effects on hADSC migration at several fluences (2.5, 5.0, and 10 J/cm2) using the Culture-Insert 2 Well in µ-Dish 35 mm, and these results will be reported in the near future.

Reviewer 4 Report

The statistical problems have been corrected.

Author Response

The authors would like to thank Reviewer #4 for his or her careful review of our manuscript and for providing us with constructive comments and suggestions to further improve the quality of our manuscript.